# An Overview of the Importance of Transition-Metal Nanoparticles in Cancer Research

**DOI:** 10.3390/ijms23126688

**Published:** 2022-06-15

**Authors:** Olga Klaudia Szewczyk, Piotr Roszczenko, Robert Czarnomysy, Anna Bielawska, Krzysztof Bielawski

**Affiliations:** 1Department of Synthesis and Technology of Drugs, Medical University of Bialystok, Kilinskiego 1, 15-089 Bialystok, Poland; robert.czarnomysy@umb.edu.pl (R.C.); krzysztof.bielawski@umb.edu.pl (K.B.); 2Department of Biotechnology, Medical University of Bialystok, Kilinskiego 1, 15-089 Bialystok, Poland; roszczenko.piotr@gmail.com (P.R.); anna.bielawska@umb.edu.pl (A.B.)

**Keywords:** gold, silver, copper, ruthenium, palladium, platinum

## Abstract

Several authorities have implied that nanotechnology has a significant future in the development of advanced cancer therapies. Nanotechnology makes it possible to simultaneously administer drug combinations and engage the immune system to fight cancer. Nanoparticles can locate metastases in different organs and deliver medications to them. Using them allows for the effective reduction of tumors with minimal toxicity to healthy tissue. Transition-metal nanoparticles, through Fenton-type or Haber–Weiss-type reactions, generate reactive oxygen species. Through oxidative stress, the particles induce cell death via different pathways. The main limitation of the particles is their toxicity. Certain factors can control toxicity, such as route of administration, size, aggregation state, surface functionalization, or oxidation state. In this review, we attempt to discuss the effects and toxicity of transition-metal nanoparticles.

## 1. Introduction

Nanoparticles are a formulation of approximately 100 nm in size [1]. Nanoparticles (NPs) have a size similar to physically existing functional units. This allows interactions with nucleic acids, proteins, or lipids. Because of this property, we observe a significant increase in the translocation of nanoparticles into internal cellular structures compared with larger particles of the same metals [2]. A remarkable advantage of nanoparticles is their ability to penetrate imperfect blood vessels created by tumor angiogenesis, which influences greater absorption of the incorporated drug by cancer cells compared with regular cells. Nanoparticles also protect the drug from inactivation and enhance the distribution properties of the drug substance. By using nanoparticles, we can increase the physicochemical parameters by significantly enhancing the solubility of the compound, which results in easier administration of the drug to the patient and higher retention time in the human body [3].

NPs based on lipid, polymer, and inorganic materials (Figure 1) are being developed with increasing accuracy. They can be optimized to deliver more personalized drugs in precision medicine. As a result of disease changes, the physiology of tissues and cells can be altered, creating heterogeneous barriers that inhibit drug penetration to the therapeutic target. Problems in the treatment of patients arise from insufficiently studied disease biochemistry as well as variability in the patient population, which affects the efficacy of nanoparticles. Since tumor-altered tissues affect the distribution of therapeutic substances, advances in controlled synthesis strategies make it possible to overcome these limitations by creating assembled structures combined with targeting particles [4]. 

Nanotechnology makes it possible to overcome the limitations of conventional treatment, such as particle distribution in the body and insufficient membrane trafficking. Nanoparticles improve the stability and solubility of compounds, facilitating transport across membranes, and increasing the drug’s residence time in the circulation, resulting in treatment efficacy [5,6]. 

Despite numerous studies on nanotechnology development, patient heterogeneity and response to treatment have limited the clinical use of nanoparticles. The few approved formulations are usually not first-line drugs, and many of them cause improvement in only a minority of patients [7]. These formulations can be part of more complex systems to target multiple biochemical pathways, maximize therapeutic efficacy against specific molecular targets, inhibit cell cycle phases, or overcome drug-resistance mechanisms. Recently, there has been a focus on generating NPs to overcome biological barriers specific to diseases or subgroups of patients, which is the core of precision medicine. Information about a particular group of patients, such as their environmental exposure, genetic profile, or co-existing diseases, is used to individualize a treatment plan [8].

Nanoparticles offer incredible possibilities in the development of disease therapies. In this review, we introduce the potential applications and limitations of metal nanotechnology in cancer treatment.

## 2. Metallic Nanoparticles

### 2.1. Gold Nanoparticles (Au NPs)

One of the most popular options for creating nanoparticles is gold. Colloidal gold was used in ancient times as a medicine for heart diseases or epilepsy. Gold nanoparticles were applied as an additive to wine, giving it a unique shade. Medieval artists used gold and silver nanoparticles for breathtaking stained glass. Au NPs are proven to have multidirectional anticancer activity. They are absorbed by the cells via endocytosis to initiate apoptosis by promoting caspase-9 expression, while nanoparticles directed to the cell nucleus promote cell cycle arrest. Owing to their physicochemical properties, they can be utilized in photothermal therapy, photodynamic therapy, or as a drug-delivery system [9].

#### 2.1.1. Mechanism of Action of Gold Nanoparticles 

Diagnostic methods should be refined to enhance the prognosis for the survival of patients with cancer. Precise and clear imaging is crucial to determine the target areas of the tumor. The popular contrast formulations are iodine-based agents, which have a short half-life in circulation. Gold nanoparticles have been receiving growing attention as a replacement for iodine-based agents. Au NPs have high biocompatibility and high X-ray absorption capacity [10]. Currently, there are two targeting pathways for gold nanoparticles—the passive way and the active way. These pathways are associated with the conjugation of the particle with tumor-specific targeting factors. Due to nanoparticles, tumors as small as a few millimeters in diameter can be detected, which is extremely important for early diagnosis [11].

An important factor influencing the activity of nanoparticles is their surface area. This dependence is inversely related to their size, which results in a large surface area for binding to a cytostatic drug, biological structure, or gene. This improves the stability and pharmacokinetic parameters of the drug. The surface-area-to-volume ratio is essential in cancer diagnostics and photothermal therapy (PTT). In PTT, light is absorbed by nanoparticles and is efficiently transformed into heat to destroy cancer cells [12]. This kind of therapy prefers smaller particles, while larger ones are recommended for cancer imaging because of their more effective light-dispersion ability. The biological response is also related to the surface area of the nanoparticles. The smaller the injected nanoparticles, the more proteins from the microenvironment can attach to them and cause a biological effect. Gold nanoparticles bind to the amino and thiol groups most effectively [13]. 

Gold nanoparticles also have inhibitory activity on tumor angiogenesis. They reduce the viability and vein formation of human umbilical vein endothelial cells (HUVECs) and the migration of human endothelial cells (HRMECs). The anti-angiogenic effect of gold in a nanometer-size formulation had a dose-dependent activity. Decreased density and permeability of blood vessels were proven in an animal model of Swiss nude mouse melanoma. Normalization of vascularization caused a reduction in tumor metastasis in mice. The molecular mechanism behind the angiogenic effect relies on the activity of vascular endothelial growth factor (VEGF) through the inhibition of signaling in various endothelial cells. Moreover, the suppression of intracellular calcium release or downregulation of proangiogenic factors have also been demonstrated [14].

Gold-based nanoparticles have been irradiated with 520 nm visible light. Studies confirmed that this treatment increased the stability of the particles. Structural analyses revealed several differences between the irradiated and non-irradiated products [15]. This formulation may prevent systemic toxicity and side effects if used as drug carriers. Covalent binding of doxorubicin is another reason for such beneficial results of Au-TAT-DOX-PEG NPs (Figure 2A). An efficacy equivalent to the cytotoxic effect of pure doxorubicin on the human osteosarcoma (HOS) cell line was reported with only 3% of the substance attached to the nanoparticle [16].

#### 2.1.2. Limitations of the Application of Gold Nanoparticles

Unmodified nanoparticles are rapidly opsonized in the blood, allowing rapid removal by macrophages. Modifying the surface is crucial to disguise particles from the immune system and enhance their uptake into the targeted tissue [17]. Functionalization of the particles modifies their optical properties, which should be taken into account in the selected direction of application. Such modification could be obtained by adsorption or covalent bonding between the components. The most common choice for functionalization is polyethylene glycol (PEG). PEGylation improves biocompatibility, and distribution in the plasma is prolonged. Nanoparticles with polyethylene glycol demonstrated no evidence of cytotoxicity against human cell lines, but accumulation has been observed in tumor-changed mouse tissues [18].

The impact of nanoparticles on the human body is not only related to their activity. The effects are also associated with the toxicity they induce and the inflammatory processes they promote. Because nanoparticles injected intravenously are used both for imaging and therapeutically, stabilizing them in plasma is essential. The stability of gold nanoparticles is required, but the stabilizers used to preserve them can also be cytotoxic. One advantageous approach is to have the particles stabilized with proteins. They have high molecular weights, defined structures, multifunctional chemical groups, and high binding affinity to metal surfaces [19]. More details are listed in Table 1.

### 2.2. Silver Nanoparticles (Ag NPs)

For thousands of years, silver was applied in various areas of human life. In medicine, the most significant uses included its antimicrobial properties. These qualities have been exploited to cure infections since ancient times. Ag NPs are absorbed in the gastrointestinal tract and skin. They circulate and accumulate in the spleen, bone marrow, brain, and heart. They are capable of passing the blood–brain barrier, then penetrating the cerebellum, and reaching the reproductive organs. Through membranes, Ag NPs migrate via the endocytosis and pinocytosis mechanisms [22]. 

With the rapid expansion of nanotechnology, silver nanoparticles have been used for antibacterial, antiviral, anticancer, and anti-inflammatory applications. Ag NPs have also become potential antifungal agents. Infections of this kind often occur in patients treated with chemotherapeutic agents. Most fungal infections are caused by *Candida* species. Silver nanoparticles show excellent antifungal activity against *C. albicans* by destroying cell membranes. The antifungal activity of silver nanoparticles is based on the formation of insoluble compounds in the fungal cell wall and disruption of the cell membrane. Damage to the cell wall and membrane leads to the release of potassium ions (K^+^). Ag NPs inhibit cellular processes that are involved in yeast budding. The chemical synthesis of nanoparticles has raised concerns about environmental pollution, so biosynthesis is being attempted. Silver nanoparticles synthesized using an aqueous extract of *Gymnema sylvestre* demonstrated significant antifungal activity against *C. albicans*, *C. tropicalis*, and *C. nonalbicans*. The silver nanoparticles were biocompatible and non-toxic to mammalian cells, and their antifungal activity was dependent on the applied concentration. It is noteworthy that Ag NPs exhibited stronger antifungal activity than antifungal drugs [23].

#### 2.2.1. Mechanism of Action of Silver Nanoparticles 

Myricetin-mediated nanoparticles were synthesized and tested for their activity on mouse embryonic fibroblasts (NIH3T3). Ag NPs induced a loss of cell viability and proliferation in a dose-dependent manner, as revealed by increased lactate dehydrogenase leakage from cells. Reactive oxygen species (ROS) were the root cause of cytotoxicity. Increased malondialdehyde amount, and decreased glutathione, ATP, and superoxide dismutase were recorded. The mitochondrial membrane potential decreased, DNA damage occurred, and p53 and p21 gene expression was upregulated. Gene ontology (GO) term analysis revealed changes in biological processes related to epigenetics following exposure to Ag NPs. This implies that nanoparticles can modulate histone gene expression [24]. 

It was reported that long-term exposure of human lung L132 and lung cancer A549 cells to Ag NPs leads to an increased expression of genes encoding antioxidant enzymes that include glutathione-S-transferases. Such enzymes are involved in the elimination of lipid peroxidation products. Ag NPs can induce cellular responses aimed at cell survival or cell death. Ag NPs induced apoptosis through the mitochondrial pathway in A549 cells, with no significant effect on L132 cells. The outcome of this study suggests that silver nanoparticles can stimulate lung-cancer cell-specific apoptosis in mammalian cells [25]. 

#### 2.2.2. Limitations of the Application of Silver Nanoparticles

Silver possesses promising properties, but the use of silver nanoparticles is limited by their instability. Ag NPs, therefore, have unrealized potential compared with the more stable Au NPs. Earlier results have shown that the physical, optical, and catalytic properties of Ag NPs are strongly dependent on their size, shape, and surface properties. This depends on reducing and stabilizing agents. Ag NPs developed for drug delivery are mostly bigger than 100 nm to accommodate the drug to be delivered. Ag NPs can also be formed into various shapes such as triangles, circles, octahedra, and polyhedra. These properties contribute to Ag NPs and have enabled their use in the pharmaceutical and biomedical engineering fields [26]. 

Toxic post-exposure outcomes of silver in humans include accumulation in the skin and eyes. Silver-doped prosthetic restorations cause blackish-blue staining in the tissues surrounding the prosthesis. These effects are reversible after cessation of exposure; elimination mainly proceeds through the liver and kidney. Reported data generated using rat liver cell line BRL 3A showed a significant decrease in the ability of cells incubated with silver nanoparticles. An alteration in mitochondrial membrane potential, a decrease in glutathione concentration, and a significant increase in reactive oxygen species concentration were revealed. Investigations were also performed on an in vitro model of alveolar macrophages, whose viability decreased directly proportionally to the nanoparticle concentration. Increased levels of ROS and pro-inflammatory cytokines such as TNF-α, IL-1, and macrophage inflammatory proteins 2 (MIP-2) were observed, indicating oxidative stress as a mechanism of action. The effect of silver nitrate nanoparticles on human periodontal tissue fibroblasts was investigated. An inverse dependence between AgNO_3_ nanoparticle size and cell viability was revealed. In contrast to AgNO_3_ NPs, Ag NPs did not exhibit reduced cell viability of periodontal tissue fibroblasts at any tested concentrations or incubation times. These results imply that silver nitrate nanoparticles between 10 and 20 nm in diameter, upon contact with oral tissue directly, depending on the concentration and exposure time, can induce a limited cytotoxic effect [27]. More details are listed in Table 2.

### 2.3. Platinum Nanoparticles (Pt NPs)

The increasing incidence of cancer among the population has challenged the development of safer treatments. Nanoparticles might offer an approach to improving the physicochemical parameters of a chemotherapeutic; moreover, they can be spontaneously biosynthesized in vivo. It has been confirmed that patients exhibit Pt NPs in their blood while receiving cisplatin. Significantly, it was reported that human serum albumin (has) is involved in the mechanism of nanoscale particle formation. Additionally, a spontaneous interaction of the particles with plasma proteins was created, leading to the generation of a protein crown. As a biocompatible drug-carrier, platinum nanoparticles can not only be tumor-targeted, but can also be formulated in vitro with a customized composition of crown proteins matched to tumor-targeting treatments. The most probable mechanism for the accumulation of Pt NPs in tumor tissue is through their interaction with HSA. Confirmed biosynthesis of the nanoparticles in the human body provides an opportunity for broader research on the use of other metals in a similar formulation. NPs in the bloodstream are usually coated with proteins; albumin spontaneously covers platinum, giving it the same targeting properties as FDA-approved albumin and paclitaxel complexes [31].

#### 2.3.1. Mechanism of Action of Platinum Nanoparticles

Platinum nanoparticles can enter and cause DNA damage in the cells, which has been proven in human colon cancer cell line HT-29. Furthermore, they exhibit catalytic and enzymatic activity, which enables them to be an effective therapeutic agent for cancer treatment. They are more effective in silencing genes and inhibiting the regeneration of tumor-transformed cells; thus, nanoparticles appear to be a safer replacement for established chemotherapeutics. The most probable mechanism of action of platinum nanoparticles is ROS generation. The addition of polyvinyl alcohol on the surface of nanoparticles guides them into cells through diffusion and by localizing them in the cytoplasm. The surface modification of nanoparticles increases their antitumor activity [32]. 

The combination of chemotherapy and photodynamic therapy (PDT) can provide more efficient cancer treatment. However, there are numerous limitations of synergistic therapy, including the uncontrolled release of the chemotherapeutic agent or limited effectiveness of PDT through hypoxia of the solid tumor [33]. The CPT-TK-HPPH/Pt nanoparticle was developed to overcome the problems of combination therapy. The particle contains a ROS-responsive prodrug composed of a thioketal bond combined with camptothecin (CPT) and photosensitizer 2-(1-hexyloxyethyl)-2-devinylpyrophenoforbide-a (HPPH). This combination enables the catalytic disintegration of hydrogen peroxide. It provides the cell with sufficient oxygen to improve the results of photodynamic therapy. It is also possible to use the synthesized particle for cancer imaging. CPT-TK-HPPH/Pt NP exhibits prolonged circulation and effective in vivo targeting of tumor tissue [34].

Electrochemical therapy (EChT) consists of inserting electrodes into the tumor to destroy cells under the influence of DC power. The cause of the effectiveness of this method is the pH modification in the area near the inserted electrodes. However, the restricted area of effect and sophisticated electrode configuration complicates the clinical application of EChT in the treatment of various tumor types. An electrically powered catalytic reaction with platinum nanoparticles may be a new approach in therapy. An electric current sets up a reaction between water molecules and chloride ions on the surface of Pt NPs, generating cytotoxic hydroxyl radicals. This mechanism, called EDT, causes tumor cell cytotoxicity within the entire electric field. Such a method is marginally invasive and can provide a homogeneous effect of lethality throughout a relatively large tumor. In one study, platinum nanoparticles were described as nanozymes with tailored catalytic activity. The study investigated the effect of an electric field on the existence of the Faraday cage effect on the surface of Pt NPs. The dissociation of water molecules with chlorine ions was demonstrated, which generated cytotoxic hydroxyl radicals (OH). According to the study, cancer cells incubated with platinum nanoparticles were able to be effectively killed with minimal invasiveness compared to EChT [35]. 

Nanoparticles are promising candidates for radiotherapy treatment. Pt NPs are remarkable in their capability to intensify the radiation dosage and have the ability to be carriers of compounds with therapeutic potential. The research data indicate that platinum nanoparticles enhance DNA damage and have excellent radiation-strengthening properties. Remarkably, these nanoparticles also possess electrocatalytic properties. Despite acceptable results in therapy, cisplatin and its analogs are not specific and promote drug resistance; thus, patients often suffer from toxic outcomes in healthy cells, which limits their clinical application. Therefore, it is crucial to search for improved and more efficient treatments to eradicate cancer [36].

#### 2.3.2. Limitations of the Application of Platinum Nanoparticles

The application of platinum nanoparticles raises safety concerns. This is caused by their interaction with the DNA of cells. As a consequence of the fact that platinum particles damage DNA, the proportion of Pt in cell nuclei is increased. The presence of platinum in the nuclear fraction of human liver cancer HepG2 and HT-29 cell lines was quantitatively analyzed after incubation with nanoparticles. In the first line, a 70% accumulation of platinum in the nucleus was observed, while in HT-29, it was only 25%. The combinatorial chemistry allowed for screening, which revealed the peptide H-Lys-[Pro-Gly-Lys]_2_-NH_2_ as the best for nanoparticle surface modification. This combination improved cytotoxicity against the liver cancer cell line, but with no impact on other cancerous and non-cancerous cell lines [37]. More details are listed in Table 3.

### 2.4. Palladium Nanoparticles (Pd NPs)

Functionalized nanoparticles can be drugs for cancer treatment, imaging, and drug carriers. Palladium and platinum NPs have the opportunity to become highly effective, with lower toxicity compared with existing conventional drugs. Their effectiveness could be enhanced by coating them with nontoxic polymers, protecting them from the immune system or inactivation in plasma. Platinum-group metal nanoparticles might replace conventional cancer drugs [39]

#### 2.4.1. Mechanism of Action of Palladium Nanoparticles

Palladium NPs exhibit high cytotoxicity. Their activity relies on a physicochemical interaction with the functional groups of cellular proteins, nitrogenous bases, and DNA phosphate groups [40]. Pd also leads to the generation of free radicals, lactate dehydrogenase leakage, and cell cycle disruption. Cell death proceeds mainly by apoptosis, autophagy, or necrosis. Pd NPs exhibit antitumor activity against the human breast cancer MCF-7 cell line. The mechanism involves the induction of apoptosis marked by the externalization of phosphatidylserine, the disintegration of the cell membrane, and the condensation of chromosomes. Further studies on these nanoparticles might support the establishment of their potential as anticancer drugs [41].

In connection with the anticancer properties exhibited by palladium, palladium-doped magnesia nanoparticles were created. Because Pd/MgO dissociates at low pH, the acidic microenvironment of tumor tissue allows the metal to exert its therapeutic effect. Pd/MgO nanoparticles exhibit antiproliferative effects against HT-29 colon and A549 lung cancer cells. These nanoparticles are sufficiently small to bypass the mononuclear phagocytic system, allowing for an extended distribution period. Magnesia nanoparticles with palladium are not selective for tumor-altered cells because they are also toxic to normal cells. Pd/MgO NPs affect cells by inhibiting replication. Thus, Pd/MgO NPs would be effective against rapidly dividing cells such as cancer cells, hepatocytes, and leukocytes, and less toxic against slowly dividing normal cells. Pd/MgO nanoparticles stimulated caspase-3 and -9 activities in HT-29 and A549 cells. A decrease in Bcl-2 and an increase in Bax protein and cytochrome C release was reported. In addition, there was an increase in the expression of the tumor suppressor protein p53 in cancer cells. Currently, the mechanism of Pd/MgO nanoparticles is considered to occur mainly through the induction of the intrinsic apoptosis pathway [42].

Theranostics is predicated on the premise that a singular platform could possess the ability to provide simultaneous therapy and diagnostics, and is considered a vehicle for targeting a variety of diseases. Nanoparticles that absorb light in the near-infrared (NIR) range have gained considerable attention due to their promising applications in cancer imaging and therapy [43]. Recently, palladium was identified as a noble metal with remarkable stability and good catalytic and mechanical properties for application in cancer therapy. Currently, this metal is applied in clinical environments for brachytherapy of prostate cancer and choroidal melanoma. There are available reports about palladium nanostructures as photothermal, anticancer, and antimicrobial therapeutics. However, the interactions of metal nanoparticles with many biological systems in organisms cause toxic effects that impede the widespread utilization of these nanostructures in new pharmaceuticals. Progress in the synthesis of nanoparticles and their surface modifications with amino acids or polymers gave them compatibility with cells and tissues, stability in physiological solutions, and simplified further modifications. The surface of the palladium nanoparticles was modified with chitosan oligosaccharide (COS). The particles were subsequently functionalized with an arginine-glycine–aspartic acid (RGD) peptide, which improved particle accumulation in the cells of breast cancer line MDA-MB-231. It resulted in an improved photothermal therapy effect under an 808 nm laser. The RGD peptide-linked COS-coated palladium nanoparticles Pd@COS-RGD (Figure 3A) exhibited high compatibility and colloidal and physiological stability [44]. 

#### 2.4.2. Limitations of the Application of Palladium Nanoparticles

Although Pd NPs are utilized in numerous catalytic processes, their application in medical fields is limited. Previous studies have revealed that palladium nanoparticles induce significant cytotoxic effects in several human cancer cell models, including respiratory, peripheral blood, ovarian, and melanoma lineage cells [45]. Efforts are ongoing to apply Pd NPs in the form of Pd sheet-coated hollow mesoporous silica NPs, as a platform for chemophotothermal cancer treatment. Palladium complexes combined with sulfone-containing polyamides exhibit antimicrobial activity, and the palladium NPs themselves modulate cytokine expression. Pd NPs are associated with toxicity in human ovarian cancer cells by enhancing oxidative stress, increasing caspase-3 activity, and inducing DNA damage [46]. More details are listed in Table 4.

### 2.5. Copper Nanoparticles (Cu NPs)

Copper is an essential mineral involved in both plant and animal metabolism. It is necessary for cross-links in connective tissue and also participates in electron-transfer catalysis in protein synthesis, cholesterol, iron, and carbohydrate metabolism. This element occurs naturally in the environment as metallic copper in the Cu^0^ form or as ionic copper Cu^1+^ or Cu^2+^. All these forms of copper induce different levels of toxicity in biological organisms. It is determined by particle size, shape, crystallinity, and aggregation. The toxic side effects of exposure to copper nanoparticles are comparable to those induced by overexposure to copper ions, but the harmful effects of nanoparticles are 10 times weaker [49]. Recent studies suggest that they can potentially be utilized in chemotherapy as a drug carrier. 

#### 2.5.1. Mechanism of Action of Copper Nanoparticles

The Hep2 and A549 cell lines exhibit poor post-exposure viability to copper oxide nanoparticles. This could be related to the release of copper ions into the medium. CuO nanoparticles induce cell death, and the kind of death could depend on the type of cell line. For instance, exposure to copper oxide nanoparticles induces apoptosis in HaCat cells, a non-cancerous cell line. As tumor cells generally are expected to avoid apoptosis to promote cell proliferation, it might explain the differences in cellular responses with HepG2 cells [50].

Tumor-initiating cells (TICs) stimulate tumor growth and metastasis. TICs significantly contribute to tumor re-growth via their intrinsic resistance to chemotherapy caused by increased DNA-repair capacity, overexpression of anti-apoptotic proteins, and improved drug efflux pumps. Copper oxide nanoparticles are cytotoxic to TIC-enriched human pancreatic cancer line PANC1. Exposure to CuO NPs reduces cell viability and ignites apoptosis in TIC-enriched PANC1 cultures to a greater extent than in standard PANC1 cultures. Cell death is associated with increased reactive oxygen species (ROS) levels and reduced mitochondrial membrane potential. Furthermore, CuO NPs inhibited tumor growth in a mouse model of pancreatic tumors. Notably, these tumors had a significantly higher number of apoptotic TICs compared with untreated mice [51]. 

Photodynamic therapy (PDT) is less invasive compared with other treatments. However, there are limitations to this treatment modality due to the restriction of light penetration of deep tissue tumors. X-ray-induced PDT may overcome these limitations. A novel sensitizer copper-cysteamine (Cu-Cy) can be activated by UV, X-rays, microwaves, and ultrasound to generate oxidative stress [52]. Cu-Cy NPs with an average size of 40 nm were screened on the melanoma cell line B16F10, because of increased cellular uptake of NPs. The particles possess a high surface area, allowing them to be efficient in generating ROS. As expected, Cu-Cy NPs were effective in inhibiting melanoma under X-ray stimulation. The results indicated that cell viability in the control group without X-ray excitation was not decreased. A dramatic decrease in cell viability was observed in a dose-dependent manner in the 2.5Gy treatment group. These findings imply poor toxicity of inactivated Cu-Cy NPs to cells. The generation of reactive oxygen species leads to tumor cell damage. Bypassing the general mechanism of action of this therapy, the in vivo immune response of Cu-Cy NPs was evaluated in mice. It was reported that Cu-Cy treatment with X-rays alone induced an increase in the number of CD4 + T and CD8 + T cells in the spleen. There were no noticeable changes in the number of macrophages, neutrophils, NK cells, and γδT cells. The combination of Cu-Cy and X-rays promoted decreased survival of B16F10 cells. Interestingly, the combination of radiation and a sensitizer promoted an anti-tumor immune response. Cu-Cy NPs may simultaneously enhance radiotherapy, chemotherapy, and immunotherapy for melanoma treatment [53].

#### 2.5.2. Limitations of the Application of Copper Nanoparticles

The pathways regulating copper metabolism are easily disrupted, which causes an oxidative damage effect. The most frequently detected is a cascade effect impairing liver function or disrupting mitochondrial respiration. An accumulation of copper particles is observed in plants and animals. The main pathway of toxicity induced by copper nanoparticles is the respiratory system. Prolonged exposure can initiate dose-dependent pneumonia and lung cell damage. Previous in vitro studies have confirmed that CuO NPs cause high toxicity in human lung epithelial cell A549, leukemia cell HL60, human breast cancer cell MCF-7, and hepatocellular carcinoma cell HepG2. Nanoparticles that have entered through the lungs also have effects on the cardiovascular system. One of the hypotheses is that inhaled particles cause inflammation and generate inflammatory cytokines that enter the circulation. Another hypothesis assumes that nanoparticles are freely released through blood vessels in the lungs into the circulatory system. This affects accumulation in specific areas of the blood vessels, which impact the cardiovascular system directly. CuO NPs demonstrate genotoxicity determined by the stimulation of inflammation and oxidative stress. Recent studies point to lysosome dysfunction as a mechanism of toxicity of copper nanoparticles. This is because of nanoparticle deposition, which leads to autophagic stress. CuO NPs also have toxic effects on vascular endothelial cells, inducing apoptosis in a caspase-independent manner [54]. Understanding copper toxicity will allow broader use of copper particles in medicine. More details are listed in Table 5.

### 2.6. Zinc Nanoparticles (Zn NPs)

Zinc is necessary for human growth and function due to its crucial role in neurological and bone metabolism and its impact on the immune system. This component also serves as a cofactor for approximately 3000 proteins and enzymes. Zinc-related proteins are involved in transcriptional regulation, apoptosis, and DNA repair [57]. Zinc ions in the cytosol are recognized as secondary signal transmitters similar to calcium ions, which are involved in the regulation of signal transduction pathways. Zinc deficiencies are related to the acceleration of the aging process; dysfunctions of the immune system, liver and kidneys; and impaired wound healing [58]. 

#### 2.6.1. Mechanism of Action of Zinc Nanoparticles

Nanometer-sized zinc oxide (ZnO NPs) has been receiving increased attention from researchers, with numerous studies indicating that the particle itself has demonstrated cytotoxic effects on cancer cells [59]. This is caused by the chemical and physical properties of zinc [60]. It is also worth noting that these nanoparticles have high selectivity toward tumor-transformed cells and the capability for a controlled release of pre-loaded cytostatic drugs. This occurs via the ability to dissolve and release Zn^2+^ ions in the acidic microenvironment of the tumor lesion. Zinc ions cause mitochondrial dysfunction and an increase in reactive oxygen species, lipid peroxidation, and DNA damage [61].

ZnO NPs exhibit cytotoxic, antifungal, and antibacterial properties. Zinc is one of the essential nutrients and nanoparticles that are recognized as safe. The stability and simplicity of production in both the processes and the equipment needed is a huge advantage. Bone sarcoma cell line MG-63 was used in one study. The cells exhibited stimulation of oxidative stress, which affected the mitochondrial membrane potential, inducing the cells into the desirable stage of apoptosis. The induction of programmed cell death significantly reduces the survival of cancer cells, while also preventing metastasis by reducing the adhesion of the MG-63 line. Rehmanniae Radix in synthesis is an example of the practical application of green-chemistry rules and environmental protection by limiting the utilization of irritating reagents [62].

ZnO NPs are distinguished from other metal nanoparticles by the properties of zinc itself. It is involved in maintaining redox balance in cells, the regulation of enzymes in hematopoiesis, and the regulation of the DNA- and protein-synthesis processes. ZnO NPs act supportively in wound treatment and exhibit anti-inflammatory properties. This is highly beneficial because of the links between prolonged inflammation and cancer development. The encapsulation of nanoparticles with polymers increases their water solubility, absorption, and transport across biological membranes, allowing the drug to be applied via the oral route, which is often more acceptable to the patient [63].

Increased accumulation in cancer-transformed cells has been demonstrated in cell lines such as A549 and adult Swiss albino mice. These mice were used to induce a carcinogenic benzo-a-pyrene-mediated tumor located in the lungs. Syringic acid-loaded zinc oxide nanoparticles were examined in both models. Combined nanoparticles had a defensive function against lung carcinogenesis. Additionally, they promoted oxidative stress in tumor cells in a concentration-dependent manner. They also disturbed the mitochondrial membrane potential and impaired adhesion, resulting in the significantly reduced viability of tumor-transformed cells [64].

Connecting zinc nanoparticles with cytostatic agents has been beneficial, as already demonstrated on the estrogen-dependent breast cancer cell line MCF-7. The authors investigated the effects of empty ZnO nanoparticles, a mixture of ZnO with doxorubicin, and ZnO NPs pre-loaded with doxorubicin. The mixture exhibited significantly higher cytotoxicity than the individual components, confirming their synergistic effects. The drug-loaded particles resulted in the highest level of anti-tumor activity. This has been attributed to improved retention and accumulation of the formulation in the targeted tissue. Therefore, drug loading inside a ZnO nanoparticle is an effective approach to fighting cancer, as it results in the specificity of targeting the toxic anticancer drug. Using this technique could minimize the risk of side effects [65].

One of the examples of improved distribution parameters is ZnO nanoparticles surrounded by a polymer. This research compared particle-loaded Nintedanib, Crizotinib, and Isotretinoin. They were stable and water-soluble, which is a remarkable advantage of this formulation. The DU145, HeLa, MCF-7, and A549 cell lines were used to perform MTT during the study. Flow cytometer studies were continued on line A549. Isotretinoin-loaded nanoparticles exhibited the most effective anti-tumor activity. These particles were more powerful than empty ZnO nanoparticles and the pure substance, confirming the synergistic activity. The surface polymer of these nanoparticles forms a capsule-like structure. Such a formulation determines the encapsulation integrity of the drug, which allows for a more effective cytotoxic effect in the targeted area. We can upload drugs with low water solubility into such a capsule. Another advantage is a greater sensitivity to pH, which is essential in the acidic microenvironment of the tumor cell [66].

#### 2.6.2. Limitations of the Application of Zinc Nanoparticles

If zinc oxide nanoparticles are to develop further, the knowledge about their mechanisms of action must be deepened. Thus far, there are not sufficient data on the effects of ZnO NPs at the cellular level. They have great potential, but we need to improve their biocompatibility with blood components. This is related to erythrocytes, the coagulation system, and an escape from the immune system that allows a time of distribution long enough to reach the therapeutic target. To provide the nanoparticles with the appropriate parameters, they were covered with a protective shell, which defended the particles from the immune system and enhanced their accumulation in the target tissue. With this formulation, we can also place particles specifically uptaken by tumor changes on the surface of the shell [67].

Many reports indicate different types of toxicity, such as genotoxicity of ZnO NPs against BIP/GRP78 cell lines; mutagenicity of these particles was also reported against Chinese hamster lung fibroblast cells. Neurotoxicity and pulmonary toxicity associated with the use of zinc particles have been confirmed. Despite this fact, many researchers claim beneficial effects of nanoparticles at lower concentrations; however, these effects vary depending on the route of administration, duration of exposure, cell lines, and types of organs or tissues [68]. More details are listed in Table 6.

### 2.7. Ruthenium Nanoparticles (Ru NPs)

Ruthenium as a metal has different oxidation states. Complexes containing Ru(III) are generally harmless to healthy tissues and are assumed to be prodrugs [69]. Conversely, Ru(II) derivatives are often cytotoxic to cancer cells. In tumor cells, ruthenium binds to and destroys DNA, thus providing a mechanism of activity that results in cell death. Like iron, ruthenium can form complexes with transferrin and albumin. To increase the anticancer properties of ruthenium, it could be combined with anticancer siRNAs to create a comprehensive compound with anticancer activity [70]. 

#### 2.7.1. Mechanism of Action of Ruthenium Nanoparticles

Investigations of the luminous and thermal properties of ruthenium are rarely conducted. The photothermal activity of ruthenium nanoparticles with a spherical structure was investigated. Compared to gold nanoparticles or carbon nanomaterials, these particles have high biocompatibility and exhibit an excellent photothermal effect. Insufficient delivery of this formulation to cells could be the main barrier that prohibits the application of nanoparticles against cancer cells. To overcome this limitation, ligands significantly enhance the ability of cells to absorb through receptor pathways, such as folic acid or peptides. Transferrin (Tf)—a serum protein that is non-toxic, immunologically neutral, and biodegradable—was the applied surface modification (Figure 4A). The plurality of transferrin receptors existing on the surface of proliferating and malignant cancer cells was confirmed. Tf-Ru NPs reached the cell via endocytosis. The efficacy of PTT treatment of Tf-Ru NPs particles on A549 line cells was tested using an MTT cytotoxicity assay. The photothermal effects on mice were also investigated. Transferrin-functionalized Ru NPs exhibited excellent target-killing ability against cancer cells. These novel nanoparticles can be used for highly effective PTT in vivo [71]. 

In one study, a ruthenium-selenium nanoparticle was prepared, and cysteine served as a linker. Se/Ru NPs were functionalized with MIL-101. The high porosity and surface modification of Se/Ru@MIL-101 protected siRNA from degradation, increased its cellular uptake, and promoted escaping from lysosomes. The purpose of the above was the silencing of MDR genes in the MCF-7/T cell line, resulting in enhanced cytotoxicity through the induction of apoptosis. The molecular mechanism of action of Se/Ru@MIL-101 included phosphorylation of p53, MAPK, and PI3K/Akt, and the disruption of mitotic spindle formation. These findings could suggest that the NP–microtubule interaction leads to a disruption of chromosome segregation and abnormal cell divisions, ultimately leading to cell cycle arrest and cell death. Indeed, this is an essential capture point, as microtubules and actin are required for transporting intracellular organelles and maintaining normal morphology during migration or cell division. Understanding the interaction between nanoparticles and microtubules could significantly affect the development of biocompatible nanomaterials for chemotherapy applications [72]. 

Transition metal sulfides (TMS) possess the capability to be beneficial for photothermal therapy of cancer. They exhibit sufficiently high absorption in the near-infrared (NIR) range. However, the rapid blood-distribution time and poor accumulation of photothermal agents inside the tumor lesion limit their application. Ruthenium sulfide-based nanoclusters (NCs) were developed to overcome these limitations. They were combined with oleic acid, then coated with denatured bovine serum albumin (DBSA) and polyethylene glycol. The obtained nanoclusters had photothermal conversion ability, exhibited anti-tumor efficacy on mouse mammary tumor cell line 4T1, and increased circulation time in the blood of mice compared with other TMS. This is attributable to an appropriate size of approximately 70 nm and a charge near 0 mV. As a result, the PEG-DBSA-RuS NC particle can prevent removal by the reticuloendothelial system (RES) and kidneys [73].

#### 2.7.2. Limitations of the Application of Ruthenium Nanoparticles

Ruthenium is a metal that has the potential for widespread use in cancer therapies alongside platinum. However, there are limitations to its utility caused by its low efficiency of delivery to its target tissues. The appearance of nanomaterials offers the possibility of targeting new substances to solid tumors via the enhanced permeability and retention effect (EPR effect) [74]. The association of nanoparticles with therapeutic substances will improve their selectivity against cancer cells. Nanoparticles shield the carried drugs from inactivation, which enhances the efficiency of the received dose of the therapeutic. This may also increase the circulation time in the blood. The functionalization of complexes with NPs can overcome their inadequacies. This is a probable future course for the development of the next generation of more effective anticancer drugs. Several Ru NPs, including ruthenium-selenium, -gold, -silicon, carbon-ruthenium nanotubes, and nano-MOF connections have been developed and screened for anticancer activity [75]. More details are listed in Table 7.

### 2.8. Titanium Nanoparticles (Ti NPs)

Titanium was discovered in 1791 in Cornwall by William Gregor and used as a paint additive. In the second part of the 20th century, the metal began to be widely used in industry, and in bone fusion and immobilization procedures or joint substitutions. Due to their corrosion resistance and biocompatibility, these materials are successfully utilized as artificial implants in orthopedic and dental surgery [78]. 

TiO_2_ exists in three crystal structures: anatase, rutile, and brookite. TiO_2_ is applied in various fields of industry based on particle size. Nanoparticles, including titanium dioxide, are transferred between the atmosphere, water, and ground as soon as they are produced. The increased usage of TiO_2_ NPs has caused worries about the potential exposure and later impact of titanium nanoparticles on the environment and human health. It is necessary to achieve a complete understanding of the effects of TiO_2_ NPs, and this may be affected by particle properties [79].

#### 2.8.1. Mechanism of Action of Titanium Nanoparticles

Titanium dioxide can enter the body orally, transdermally, or intravenously. The existing data indicate no or minimal gastrointestinal permeation of titanium nanoparticles. Studies with pharmacokinetic modeling revealed that high levels of TiO_2_ NPs can lead to agglomerate formation. This results in an increase in uptake by macrophages. In a study performed on rats, a dosage of 5 mg of TiO_2_ NPs per kg body weight was applied intravenously and the models were observed for 28 days. The subjects were healthy before the study. Histopathological examination showed no accumulation at detectable levels in the blood, plasma, or brain cells. The highest concentration of nanoparticles was found in the liver. Lower concentrations were observed in the spleen, lungs, and kidneys. Titanium dioxide nanoparticles are eliminated from the body at a low rate, which suggests tissue accumulation. This issue is not severe with human PDT because a photosensitizer is administered. In addition, the study on rats showed that the level of TiO_2_ NPs in urine was higher than in feces, which suggests renal excretion as the primary route of elimination of TiO_2_ nanoparticles [80].

Permeability research on TiO_2_ NPs revealed the lack of penetration of these nanoparticles in both intact and damaged skin. NPs are localized in the epidermal but not in the dermal layer. The concentration in the skin was similar in both models. The cause could be strong stability or weak ionization. The absence of the permeation of titanium nanoparticles in both intact and damaged skin and the minimal cytotoxicity exhibited on human keratinocytes (HaCaT) implies the poor toxic potential of these nanoparticles at the dermal level [81].

Bulky titanium dioxide particles are biologically neutral and used as pigments, food colorants, cosmetics, and sunscreens. However, it is notable that TiO_2_ NPs exhibit cytotoxic properties. When exposed to UV radiation, TiO_2_ NPs generated hydroxyl radicals, which caused oxidative stress in Chinese hamster ovary cells. Furthermore, titanium dioxide nanoparticles induced oxidative damage in human bronchial epithelial cells (16HBE14o-). In mouse macrophage cell line RAW 264.7, induced ROS generation was followed by ERK 1/2 activation. This led to the secretion of pro-inflammatory cytokines such as tumor necrosis factor TNF-α and MIP-2. Additionally, p53 protein accumulation was observed. All these studies suggest that oxidative stress is the main mechanism of cytotoxicity caused by titanium dioxide nanoparticles. TiO_2_ NPs are pre-irradiated with UV-induced DNA damage in human Hep G2 liver cancer cells. The exposure of cancer cells to titanium nanoparticles reduced cell viability, and the EGFR receptor and its associated kinases became dephosphorylated. The dysregulated interaction between EGFR and its ligand (EGF) leads to a reduction in proliferation and viability signals and, terminally, to apoptosis. Because EGFR is present in healthy tissues, albeit to a lower degree compared to cancer cells, the outcomes could relate to the cytotoxic properties of TiO_2_ NPs in non-tumorigenic cells. However, as titanium dioxide nanoparticles are also known to have cytotoxic effects by inducing oxidative stress, their cytotoxicity may be either caused by ROS or disruption of the EGFR pathway [82].

#### 2.8.2. Limitations of the Application of Titanium Nanoparticles

Nanoparticles are possibly poisonous to cells because of their size and associated reactivity. They can move through cell membranes and cause damage to either proteins or DNA through ROS generation. Metal oxide nanoparticles can generate radicals when activated with light energy. TiO_2_ nanoparticles can interact with UV radiation in the environment to cause damage to biological hosts, while they can be neutral to the same organisms in the absence of light. The kinetics of TiO_2_-activated NPs is fast, but decreases significantly when light exposure is stopped. Under minimal or no UV exposure, titanium oxide nanoparticles cause mortality, and decrease growth and the negative impacts on cells and the DNA of organisms [83]. More details are listed in Table 8.

### 2.9. Vanadium Nanoparticles (V NPs)

Vanadium is a trace element that exhibits insulinomimetic, antiparasitic, anticancer, and osteogenic activities. Vanadium-based derivatives are successors to platinum complexes. Regrettably, the usability of vanadium compounds is restricted by the toxicity reported in animals and cell lines. New targeting strategies must be formulated to deliver the complexes to selected tissues. One of these technologies includes the use of nanoparticles to improve distribution parameters [87]. Different surface modifications of nanoparticles exhibit various properties and variable accumulation in organs such as bone marrow, spleen, lung, or liver. In the present study, nanostructured lipid carriers (NLCs) and polymeric nanoparticles were used for the surface modification of V NPs. NLCs reveal stability in the storage of loaded drugs, prolonged half-life in the bloodstream, and enhanced biocompatibility. They tend to be effective and safe in parenteral, oral, ocular, and inhaled applications. However, this delivery system is also associated with high release rates as a result of nanoparticle erosion in the body. Accumulation in the liver and spleen results in potential side effects associated with these structures. The modification of nanoparticles with polymers is an alternative. These can enhance the delivery of drugs, proteins, and antigens via different routes of administration. Furthermore, polymeric nanoparticles tend to exhibit low cytotoxicity and good compatibility with cells and tissues in vivo [88].

#### 2.9.1. Mechanism of Action of Vanadium Nanoparticles

Transition metals such as copper, iron, chromium, and vanadium generate ROS through Fenton-type or Haber–Weiss-type reactions. The nanoparticles of these metals induce oxidative stress caused by their small size and more reactive surface area, leading to cellular responses. It is worth noting that vanadium nanoparticles possess enzymatic activity. However, the unquestioned advantages of this formulation are restricted by reduced cellular uptake, resulting in decreased biological activity. Studies performed on the MDA-MB-231 cell line may lead to an understanding of the mechanism of molecular interactions with cellular organelles, which would also explain their toxicity. V NPs acted as an initial suppressor of reactive oxygen species; however, the antioxidant activity decreased after 24 h, possibly caused by the loss of reactivity associated with surface saturation. They mainly accumulated in both lysosomes and mitochondria, impairing their physiological functions, thereby promoting autophagy. Upon prolonged exposure, vanadium nanoparticles induced cell cycle arrest and inhibited cell migration. Antioxidant activity decreased after 24 h, which may be related to the loss of surface reactivity associated with surface saturation. After 48 h, the acidic environment in lysosomes led to the dissolution of vanadium nanoparticles to vanadate ions, which participate in a Fenton-type reaction. This caused the generation of reactive oxygen species. These ions can pass through the membranes of organelles and further enhance the concentration of intracellular free radicals (Figure 5). The extended uptake of V NPs could mobilize the cellular defense machinery, thereby raising the level of intracellular ROS even more significantly [89].

V_2_O_5_ NPs might provide a new line of approach for novel therapies for melanoma and other cancers via their antiangiogenic properties. Research conducted with V_2_O_5_ NPs revealed an effective inhibition of proliferation of B16F10, A549, and PANC1 cell lines without affecting the CHO, HEK-293, and NRK-49F cell lines. There was inhibition of VEGF-induced endothelial cell migration of HUVECs and EA. Impaired angiogenesis in a chick embryo model was also shown. The mechanism of action of V_2_O_5_ NPs is based on the generation of intracellular reactive oxygen species, which subsequently increase the upregulation of p53 protein, resulting in the apoptosis pathway. Moreover, the administration of V_2_O_5_ NPs to C57BL6/J mice with melanoma significantly increased their survival compared to potential new substances targeting melanoma. The subchronic toxicity results of C57BL6/J mice elucidated no significant toxic effects of titanium pentoxide nanoparticles, which indicates their biocompatibility [90].

#### 2.9.2. Limitations of the Application of Vanadium Nanoparticles

Vanadium dioxide nanoparticles have been investigated and applied in various fields, resulting in concerns about the safety of exposure to VO_2_ NPs. The effects of VO_2_ NPs (N-VO_2_) and titanium dioxide-coated VO_2_ NPs (T-VO_2_) were investigated against human lung cell lines A549 and BEAS-2B. Both variants of VO_2_ NPs induced dose-dependent cytotoxicity, and both cell lines exhibited similar sensitivity to these nanoparticles. Under the same conditions, T-VO_2_ NPs showed slightly less activity than N-VO_2_ in both cells, indicating that surface coating with TiO_2_ reduced the toxicity of the formulation. This coating further affected the surface properties of the VO_2_ NPs and consequently reduced vanadium release. The induced loss of cell viability was attributed to the inhibition of proliferation and entrance to the apoptosis pathway, as evidenced by mitochondrial membrane damage and increased caspase-3 levels. A primary mechanism of VO_2_ NP cytotoxicity was oxidative stress and less reduced glutathione in A549 and BEAS-2B cells [91].

### 2.10. Iron Nanoparticles (I NPs)

Iron deficiency affects approximately one-third of the population and is a root cause of anemia. Iron absorption is carried out in the duodenum and is regulated by the divalent metal transporter. Iron reaches the circulation through ferroportin and is bound to transferrin. Intracellularly, iron is linked to ferritin to avoid cell damage from the generation of oxygen free radicals. Total-body iron levels are precisely regulated because there are no mechanisms to excrete iron from the body other than through blood loss or cell turnover. Excess amounts of iron and inflammation stimulate the production of hepcidin, which blocks the absorption of iron into the circulation by degrading ferroportin and inhibits the release of stored iron [92].

#### 2.10.1. Mechanism of Action of Iron Nanoparticles

Iron oxide nanoparticles (IO NPs) are biocompatible, stable, and environmentally safe. Owing to these properties, they are a promising platform in biomedical applications. They can bind tumor-targeting ligands such as monoclonal antibodies, peptides, or small molecules for diagnostic imaging. The magnetic properties of iron oxide nanoparticles can be magnetic vectors that can use a magnetic field gradient and magnetic contrast agents in magnetic resonance imaging (MRI). They can also be an agent for hyperthermia or thermoablation under the influence of a high-frequency magnetic field [93].

Due to the properties of nanoparticles and their ability to attach ligands to them, they can be developed with targeted drug-delivery for cancer treatment and diagnosis. Thus, it was possible to obtain conjugates of IO NPs coated with polymers bearing folic acid to obtain better anti-cancer properties [94], where doxorubicin was then loaded inside. The final formulation exhibited a faster release of doxorubicin at a lower pH in the tumor microenvironment. This is a beneficial factor in targeted therapy for cancer treatment. The activity of this formulation was evaluated on MCF-7 cells and xenograft MCF-7 breast tumors of nude mice in vivo. Tests were carried out using magnetic resonance imaging (MRI) [95].

Since almost all nanoparticles administered intravenously are eliminated by macrophages in the liver, spleen, bone marrow, or lymph nodes, they may find applications for the visualization of tumors and metastases in these organs. Iron oxide nanoparticles, particularly in the 50–150 nm range, are used, relatively, for the observation of primary tumor lesions in the liver. Due to their high hepatic uptake in healthy liver tissues compared to tumor-lesioned tissues, this uptake is reduced by a reduced number of macrophages. MRI enhanced with IO NPs is used to differentiate between malignant liver-lesions, benign neoplastic lesions, or hyperplastic liver-lesions. Currently, ferumoxsil (Lumirem, Guerbet, NJ, USA; GastroMARK, Mallinckrodt Pharmaceuticals, Ireland, EU) is the only iron oxide nanoparticle for imaging purposes and is administered as an oral contrast enhancer in gastrointestinal diagnostics [96].

Ultrasmall superparamagnetic iron oxide nanoparticles (20–30 nm) can be used to assess lymph node status in cancer classification and metastasis. Through their property of extended circulation time in the bloodstream, they penetrate tissues; from there, they are removed by lymphatic vessels, ultimately ending up in the macrophages located in lymph nodes. Metastases in the lymph nodes impair their function and reduce the number of macrophages, which manifests as a lower uptake of iron oxide nanoparticles and a lower signal in MRI imaging [97].

Additionally, it is possible to obtain iron nanoparticles from living organisms such as plants, animals, or bacteria. The in vitro anticancer activity of iron oxide nanoparticles synthesized with biological substrates was evaluated. Green synthesis using an aqueous extract from the seaweed *Sargassum muticum* produced a formulation that could inhibit the growth of breast cancer (MCF-7), cervical cancer (HeLa), liver cancer (HepG2), and leukemia (Jurkat) cells, with an apoptotic response, wherein no toxicity was observed on normal cells [98]. The possibility of obtaining iron oxide nanoparticles using a newly extracted bacterial supernatant from *Bacillus cereus* that showed cytotoxicity on MCF-7 breast cancer cells (IC50 > 5 mg/mL) and mouse embryonic fibroblasts (IC50 > 7.5 mg/mL) [99].

#### 2.10.2. Limitations of the Application of Iron Nanoparticles

Among the reported toxic effects of IO NPs, several have been demonstrated on specific cell types, for example, cardiomyocytes. The main organs exhibiting physiological dysfunction after exposure to IO NPs depend on the route of administration, and are mainly the liver, spleen, and lungs. Specific organ toxicity may be related to the development of oxidative stress, which leads to increased concentrations of ROS and MDA, as well as abnormal amounts of GSH, NADH, and SOD. The use of polymer or lipid coatings reduces the risk of toxic effects [100]. More details are listed in Table 9.

## 3. Conclusions

Metallic nanoparticles are one of the main strands of research to cure malignant changes. Nanocarriers can attach ligands specific to receptors that are overexpressed in cancer cells. NPs exhibit significantly more favorable physicochemical properties compared with their larger equivalents. They often have lower toxicity, are more effective reaction catalysts, and have more effective pharmacological effects. Metal nanoparticles offer opportunities for their use in cell imaging as biosensors and drug-delivery systems. This is related to their small size and reactive surface area [89]. Due to their multifunctionality, nanotechnology-based imaging and therapy could be a breakthrough in cancer research [103]. Depending on the selected application, the appropriate metal to formulate the nanoparticles should be selected. Nowadays, gold, silver, platinum, and zinc nanoparticles are well investigated for drug delivery. Titanium nanoparticles and palladium nanoparticles also demonstrate promising properties in this domain. Iron and gold nanoparticles might provide a promising approach for cancer imaging. The luminous and thermal properties of ruthenium, platinum, and copper nanoparticles should be considered in PTT and PDT therapies. The main limitation of nanoparticles is their toxicity, which can be controlled by the route of administration or by surface modification, by coating them with biodegradable polymers.

## Figures and Tables

**Figure 1 ijms-23-06688-f001:**
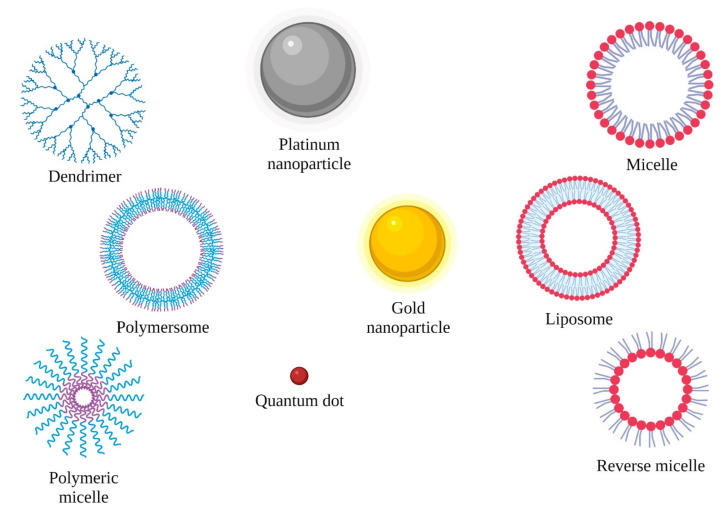
Types of nanoparticles (created using BioRender.com, access date: 6 May 2022).

**Figure 2 ijms-23-06688-f002:**
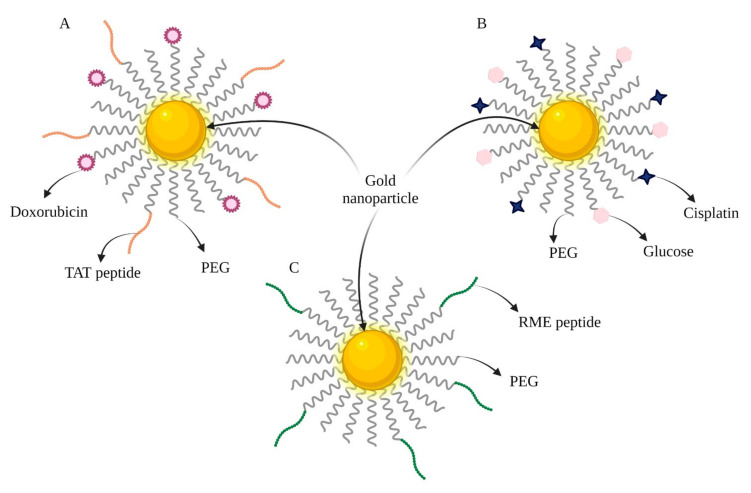
Combined nanoparticles: (**A**) Au-TAT-DOX-PEG; (**B**) CG- Au NPs; (**C**) Au NPs-RME (created using BioRender.com, access date: 6 May 2022).

**Figure 3 ijms-23-06688-f003:**
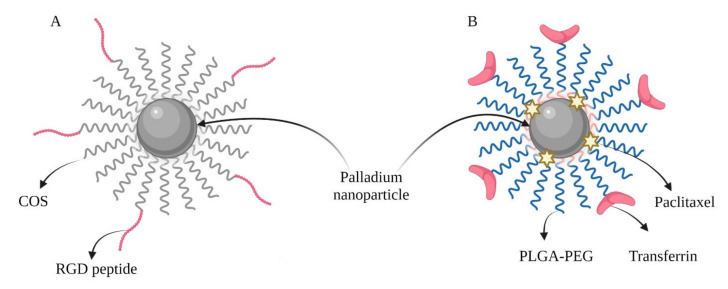
Combined nanoparticles: (**A**) Pd@COS-RGD; (**B**) PdNP/PTX-loaded PLGA-PEG NPs. (created using BioRender.com, access date: 6 May 2022).

**Figure 4 ijms-23-06688-f004:**
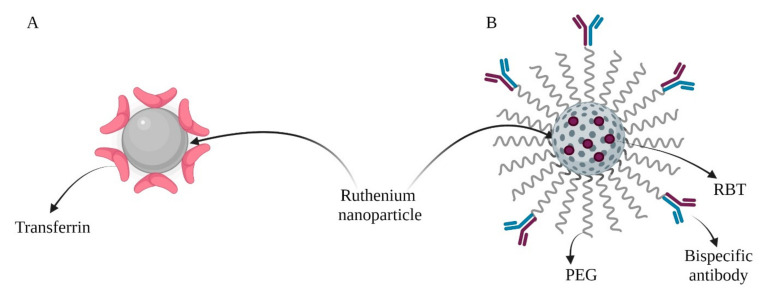
Combined nanoparticles: (**A**) Tf-Ru NPs; (**B**) HMRu NPs, (created using BioRender.com, access date: 6 May 2022).

**Figure 5 ijms-23-06688-f005:**
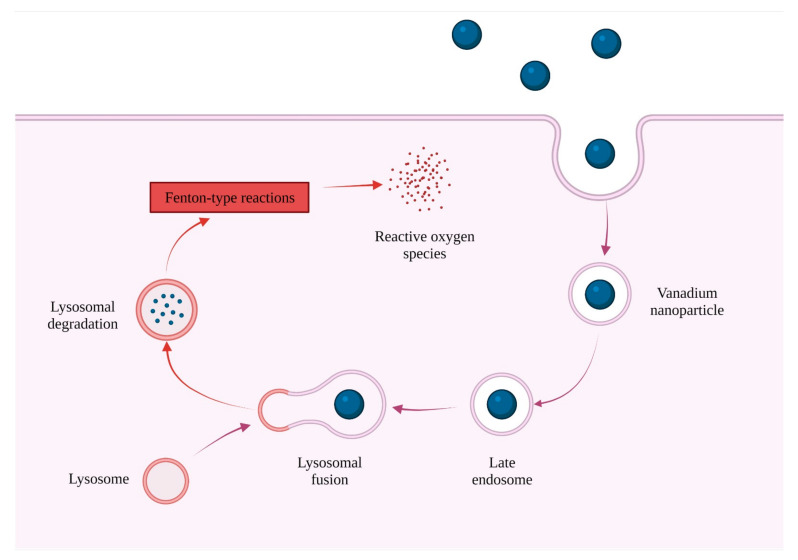
Interaction of vanadium nanoparticles with lysosomes (created using BioRender.com).

**Table 1 ijms-23-06688-t001:** Selected connections of biologically active molecules with gold nanoparticles.

Therapy	Connected Particles	Experimental Model	Molecular Mechanism	Reference
CHT/RT	Cisplatin, Glucose (Figure 2B)	A431 cells Mice bearing A431 tumor	Simultaneously acts as a radiosensitizer, drug carrier, and tumor imaging agent.	[20]
RT	An adenoviral receptor-mediated endocytosis (RME) peptide (Figure 2C)	MDA-MB-231 cells NOD/SCID mice bearing MDA-MB-231 tumor	Acts as a radiosensitizer.	[21]
CHT	Doxorubicin (DOX) TAT peptide	HOS and NHDF cell lines	Cytotoxic effect of pure doxorubicin with 3% of the substance incorporated into the nanoparticles.	[16]

**Table 2 ijms-23-06688-t002:** Selected connections of biologically active molecules with silver nanoparticles.

Therapy	Connected Particles	Experimental Model	Molecular Mechanism	Reference
CHT	Camptothecin (CPT)	HeLa cells	CPT and Ag NPs caused cell death by inducing a mitochondrial-membrane permeability change and activation of caspase 9, 6, and 3.	[28]
CHT	Paclitaxel (PTX)	MDA-MB-231, MCF-7, 4T1, Saos-2, and HUVEC cells	Ag NPs-PTX reduced the PTX dose significantly, which may prevent serious side effects.	[29]
CHT	Capecitabine	MCF-7	Lower doses of capecitabine bonded with Ag NPs can reduce unwanted side effects.	[30]

**Table 3 ijms-23-06688-t003:** Selected connections of biologically active molecules with platinum nanoparticles.

Therapy	Connected Particles	Experimental Model	Molecular Mechanism	Reference
PDT/CHT	Camptothecin (CPT) 2-(1-hexyloxyethyl)-2-divinyl pyropheophorbide-a (HPPH)	CT26 cells CT26 tumor-bearing BALB/c mice	Pt NPs could decompose H_2_O_2_ into oxygen, leading to improvement in the ROS-generation ability of HPPH. The fluorogenic nature of HPPH enabled the visualization of the color cellular uptake in vitro and tissue distribution in vivo via fluorescence imaging and photoacoustic imaging.	[34]
CHT	H-Lys-[Pro-Gly- Lys]_2_-NH_2_	HepG2, MCF-7, HeLa, PC3, A431, A549, A2780 and HT-29 cell lines	The combination of high cellular uptake and an oxidative environment was the reason that peptide-coated Pt NPs had the highest cytotoxicity, combined with selectivity, for hepatic cancer cells.	[37]
CHT/PTT	Doxorubicin hydrochloride (Dox)	MCF-7/ADR cells	Pt NPs had a high loading capacity for chemotherapeutic drugs and could deliver Dox into tumor cells. Moreover, they had great photothermal conversion capabilities and photostability.	[38]

**Table 4 ijms-23-06688-t004:** Selected connections of biologically active molecules with palladium nanoparticles.

Therapy	Connected Particles	Experimental Model	Molecular Mechanism	Reference
CHT/PDT	Paclitaxel (PTX) Transferrin (Tf) (Figure 3B)	MDA-MB-231 and MCF-7 cells MCF-7 tumor-bearing BALB/c nude mice	Synergistic anticancer effect in combination with PTX. Transferrin on the surface improved their uptake into cancer cells.	[47]
PTT	RGD (arginine-glycine-aspartic acid) Chitosan oligosaccharide (COS)	MDA-MB-231, HEK-293, MG-63 cells BALB/c nude mice bearing MDA-MB-231 tumor	Chitosan oligosaccharide (COS) improved biocompatibility. Particles were functionalized with RGD peptide, which improved their accumulation.	[44]
CHT	Tubastatin-A	MDA-MB-231	TUB-A and Pd NPs synergistically induced apoptosis by decreasing cell viability and inhibiting HDAC activity. This effect was more significant in cytotoxicity, loss of mitochondrial membrane potential and increases in caspase-3 activity, DNA fragmentation, and expression of proapoptotic genes.	[48]

**Table 5 ijms-23-06688-t005:** Selected connections of biologically active molecules with copper nanoparticles.

Therapy	Connected Particles	Experimental Model	Molecular Mechanism	Reference
**PDT**	Cysteamine	HepG2 cells, nude mice bearing HepG2 tumor; B16F10 cell line, nude mice bearing B16F10 tumor	Generation of substantial ROS levels, induction of an antitumor immune response. The presence of oxygen led to the creation of the photodynamic reaction.	[52,53]
**CHT**	Triethylene- tetramine-bis(dithiocarbamate) (TETA-DTC), poly-L-histidine (PHis), Arg- Gly-Asp peptide (RGD)-conjugated poly(ethylene glycol) (RGD-PEG)	HUVECs, BEAS-2B, MCF-7, MDA-MB-231, and 4T1 cells Female BALB/c mice bearing 4T1 tumor	Suppression of angiogenesis through RPTDH-induced copper deficiency and stimulation of antitumor immunity in vivo	[55]
**PTT**	Quaternized chitosan (QCS)	4T1 cell lines 4T1 tumor-bearing mice	Under NIR light, Cu ions from the CuS nanoparticles led redox reactions to generate ROS production, stimulating inflammation and initiating proapoptotic cellular signaling.	[56]

**Table 6 ijms-23-06688-t006:** Selected connections of biologically active molecules with zinc nanoparticles.

Therapy	Connected Particles	Experimental Model	Molecular Mechanism	Reference
CHT	Quercetin (Q) PBA acid (3-carboxybenzeneboronic acid)	MCF-7 and MCF-10a Ehrlich’s ascites carcinoma, solid tumor-bearing male Swiss albino mice	Induction of cytotoxic effect via ROS enhancement effects of ZnO and free Q in cancer cells. Nanoparticles did not show systemic toxicity in tumor-bearing mice and were found to reduce tumor-associated toxicity in the liver, kidney, and spleen.	[60]
CHT	Syringic acid	A549 cell line Male adult swiss albino mouse model of induced lung cancer	Moderate ROS generation, disrupted mitochondrial membrane potential, morphological modification by dual staining and viability, and non-viability by cell adhesion assay.	[64]
CHT	Isotretinoin Nintedanib Crizotinib	DU145, HeLa, MCF-7, and A549 cell lines	The loading capacity of the capsules was higher than on NP surfaces. The pH sensitivity of ZnO−ISO was also higher.	[66]

**Table 7 ijms-23-06688-t007:** Selected connections of biologically active molecules with ruthenium nanoparticles.

Therapy	Connected Particles	Experimental Model	Molecular Mechanism	Reference
PTT	Transferrin	A549 and HEK-293 cell lines Male mice bearing A549 tumor	Acts as a photothermal agent. High absorption under NIR irradiation and efficient heat transformation for photothermal therapy.	[71]
PTT/PDT/IMT	Bispecific antibodies (SS-Fc) Fluorescent anti-tumor complex ([Ru(bpy)2(tip)]2+, RBT (Figure 4B)	HIEC-6 cells, Caco-2, SW480, HCT116, CT26.WT cell lines Female BALB/c mice bearing CT26-CEA cells	Nanoparticles delivered RBT to solid tumors for combined HMRu-based PTT, RBT-induced PDT, and SS-Fc-mediated immunotherapy.	[76]
Starvation therapy and Oxidation therapy	Glucose oxidase (GOx)	4T1 and U87 cell lines BALB/c nude mice with 4T1 xenograft tumors	Compound converted H_2_O_2_ to toxic ^1^O_2_, thereby inducing tumor cell apoptosis and also catalyzing the conversion of H_2_O_2_ to O_2._	[77]

**Table 8 ijms-23-06688-t008:** Selected connections of biologically active molecules with titanium nanoparticles.

Therapy	Connected Particles	Experimental Model	Molecular Mechanism	Reference
CHT	Erlotinib (ERL) and Vorinostat (SAHA)	WISH, MDA-MB-231, and MCF-7 cell lines	Increase in total apoptosis in all treatments. ERL- and SAHA-loaded TiO_2_ NP treatments arrested cells at the G2/M phase. PLAB2 was upregulated in ERL- and SAHA-loaded TiO_2_ NPs compared with control cells.	[84]
CHT	Doxorubicin (DOX)	MCF-7 and MCF-7/ADM cells	DOX can be released from the surface of TiO_2_ nanoparticles in the acidic environment of endosomes or lysosomes.	[85]
PTT, SDT	[Ir(2-phenylbenzo[d]thiazole)2(4-(1-phenyl-1H- imidazo [4,5-f][1,10]phenanthrolin-2-yl)benzoic dopamine amide)]Cl	HeLa cells HeLa tumor-bearing mice	Localized and accumulated in cancerous over non-cancerous cells. Upon irradiation in the near-infrared- II region at 1064 nm or ultrasound radiation and their combination, acted as an imaging agent and as a therapeutic agent.	[86]

**Table 9 ijms-23-06688-t009:** Selected connections of biologically active molecules with iron nanoparticles.

Therapy	Connected Particles	Experimental Model	Molecular Mechanism	Reference
CHT	Doxorubicin (DOX)	CHO and HFLF cells	Redox-responsive properties resulted from a disulfide bond; the rapid release was observed in intracellular reducing potential.	[95]
CHT	5-Fluorouracil (5-FU)	Female athymic nude mice bearing HT-29 tumor	DNA damage and increased stress levels in cells, impact on second messenger and nuclear receptor signaling, caveolar-mediated endocytosis with DAMPs.	[101]
Immunotherapy	d,l-lysine M75 monoclonal antibody	B16 mouse melanoma cells, C33a human cervical cancer cells	Antibody-conjugated nanoparticles can target malignant cells and accumulate in the cytoplasm.	[102]

## Data Availability

Not applicable.

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
