# Peer review of "An Overview of the Importance of Transition-Metal Nanoparticles in Cancer Research"

_ijms, 2022, doi:10.3390/ijms23126688_

Round 1
Reviewer 1 Report
This manuscript attempts to give an overview of the potential applications and limitations of metal based nanoparticles in cancer treatments. The transition metal nanoparticles examined within the review include gold, silver, platinum, palladium, copper, zinc, ruthenium, titanium, vanadium and iron
Although this is a fairly good attempt to present the efforts that have been made so far with these metals in the nanotechnology setting, I think that this work needs major revision and is not suitable for publication in IJMS in its current form.
The manuscript lacks focus and the transition between paragraphs is hazy to the point that confuses the readership. It is not clear what exactly is reviewed in the manuscript. Is it the formation of nanoparticles, the mode of action, the linkage of bioactive molecules, their use in various treatment approaches? At the end of each section, the reader is not sure which aspects were reviewed. The assembly of information appears to be chaotic lacking a solid, well defined structure.
Additionally the paper is poorly written. It should be edited for correct English in terms of style, correct words use, and brevity

Author Response
Response to the Reviewer 1:
According to the Reviewer’s suggestions:
The manuscript lacks focus and the transition between paragraphs is hazy to the point that confuses the readership. It is not clear what exactly is reviewed in the manuscript. Is it the formation of nanoparticles, the mode of action, the linkage of bioactive molecules, their use in various treatment approaches? At the end of each section, the reader is not sure which aspects were reviewed. The assembly of information appears to be chaotic lacking a solid, well defined structure.
We rearranged the paper to be more organized. We have divided the text into specific subsections. The changes in the layout of the paper are marked in red in the text. The whole review was to perform profiling of nanoparticles in terms of their advantages (ability to combine them with active molecules, molecular mechanism) and disadvantages (toxicity).
Additionally, the paper is poorly written. It should be edited for correct English in terms of style, correct words use, and brevity
As suggested, we have done the language revision, a certificate is attached.
We would like to thank the Reviewer for the valuable comments and suggestions. Accordingly, we have revised and tried our best to improve the manuscript. We sincerely hope that the revised manuscript will meet your approval.

Reviewer 2 Report
The review entitled "An overview of the importance of transition metal nanoparticles in cancer research" contains scientific data on the problems and possibilities of using metal nanoparticles for cancer treatment, which is of scientific interest. The authors have done a lot of work by presenting data on ten different metals, making comparisons and generalizations. The article is written in excellent language, the narrative is consistent, the text is logical and structured. The illustrations are made qualitatively and clearly. The number of references to literary sources is quite large. The conclusion is brief, but quite succinct. It is proposed to accept the article after minor changes.
1) Table 2 is separated by an offset to the second page.
2) Table 3 is separated by an offset to the second page.
3) In Chapter 2.2, you can add information about the fungicidal effect of silver nanoparticles, for example, https://doi.org/10.3390/mi12121480 , in line 149.
4) There is a lot of unfilled free space on page 9.
5) There is too much free space between Figure 3 and the signature.
6) Line 334. The title and the text of the chapter are on different pages.
7) Table 6 is separated by an offset to the second page.
8) There is too much free space between Figure 4 and the signature.
9) Table 7 is separated by an offset to the second page.
10) Line 616. The title and the text of the chapter are on different pages.
11) Line 746. Extra line.
12) It is proposed to expand chapter 3 Conclusions a little. It is desirable to summarize such valuable information given in the previous chapters at the end of the article. For example, you can list the most and least effective metals, from the point of view of the authors. To list the main advantages of metal nanoparticles due to the mechanisms of action on membranes and so on. List the main limitations, at least briefly.
Of course, minor problems with the design can not spoil the positive impression of an excellently written article.
Author Response
Response to the Reviewer 2:
The authors would like to thank the Reviewer for taking time to review our paper. According to the Reviewer’s suggestions:
Table 2, Table 3, Table 6, and Table 7 are separated by an offset to the second page.
We verified that all tables are not split.
In Chapter 2.2, you can add information about the fungicidal effect of silver nanoparticles, for example, https://doi.org/10.3390/mi12121480
We have included a suggested reference to Chapter 2.2
There is too much free space between Figure 3/Figure 4 and the signature.
We made the figures shorter so that there was less free space.
Line 334/ Line 616. The title and the text of the chapter are on different pages.
We verified that all titles and text are on the same pages.
It is proposed to expand chapter 3 Conclusions a little. It is desirable to summarize such valuable information given in the previous chapters at the end of the article. For example, you can list the most and least effective metals, from the point of view of the authors. To list the main advantages of metal nanoparticles due to the mechanisms of action on membranes and so on. List the main limitations, at least briefly.
As suggested by the Reviewer, we expanded chapter 3.
The authors would like to thank the Reviewer for the thorough analysis of the paper and the valuable comments that significantly increased the scientific value of the article. We hope that the revised paper will meet your approval.
